# Frailty Network in an Acute Care Setting: The New Perspective for Frail Older People

**DOI:** 10.3390/diagnostics12051228

**Published:** 2022-05-13

**Authors:** Sara Salini, Silvia Giovannini, Marcello Covino, Christian Barillaro, Nicola Acampora, Ester Manes Gravina, Claudia Loreti, Francesco Paolo Damiano, Francesco Franceschi, Andrea Russo

**Affiliations:** 1Department of Aging, Neurological, Orthopaedic and Head-Neck Sciences, Fondazione Policlinico Universitario A. Gemelli IRCCS, 00168 Rome, Italy; christian.barillaro@policlinicogemelli.it (C.B.); nicola.acampora@policlinicogemelli.it (N.A.); ester.mansesgraavina@policlinicogemelli.it (E.M.G.); claudia.loreti@policlinicogemelli.it (C.L.); andrea.russo1@policlinicogemelli.it (A.R.); 2Department of Geriatrics and Orthopaedics, Università Cattolica del Sacro Cuore, 00168 Rome, Italy; fp.damiano@gmail.com; 3UOS Riabilitazione Post-Acuzie, Fondazione Policlinico Universitario A. Gemelli IRCCS, 00168 Rome, Italy; 4Emergency Department, Fondazione Policlinico Universitario A. Gemelli IRCCS, 00168 Rome, Italy; marcello.covino@policlinicogemelli.it (M.C.); francesco.franceschi@policlinicogemelli.it (F.F.); 5Department of Translational Medicine and Surgery, Università Cattolica del Sacro Cuore, 00168 Rome, Italy

**Keywords:** Frailty Unit, multidimensional evaluation, Emergency Department, older people, discharge plan

## Abstract

The incidence of elderly patients who come to the emergency room is progressively increasing. The specialization of the physician units might not be adequate for the evaluation of this complexity. The present study aimed to present a standard procedure, called ‘The Geriatric Frailty Network’, operating at the Policlinico Gemelli IRCCS Foundation, which is configured specifically for the level II assessment of frail elderly patients. This was a retrospective study in 1191 patients aged over 65, who were evaluated by the Geriatric Frailty Unit directly after emergency department admission for one year. All patients underwent multidimensional geriatric evaluation. Data were collected on demographics, co-morbidity, disease severity, and Clinical Frailty Scale. Among all patients, 723 were discharged directly from the emergency room with early identification of continuity of care path. Globally, 468 patients were hospitalized with an early assessment of frailty that facilitated the discharge process. The geriatric frailty network model aims to assist the emergency room and ward doctor in the prevention of the most common geriatric syndromes and reduce the number of incongruous hospitalizations.

## 1. Introduction

In recent decades, demographic changes, the increase in life expectancy, and the progressive aging of the population have had significant repercussions on the composition of the patient population accessing the hospital, with a consequent necessity to reorganize the whole care process [1,2]. Moreover, the incidence of elderly patients (aged >65) who come to the emergency room every day is also progressively increasing [3,4,5], especially in a pandemic era [6,7].

The specialization and the sectoral experience of a single hospital unit might not be adequate for the evaluation of this complexity, making it necessary, in some cases, to be assisted by a consultant expert in the multidimensional assessment of this patient category. The global care of these patients through the geriatric evaluation favors better outcomes in terms of in-hospital and out-of-hospital mortality [8,9], the morbidity linked to hospitalization (e.g., delirium, hypomobility syndrome, malnutrition, bedsores, etc.), length of hospitalization, re-hospitalizations, etc. [10,11,12].

In this general situation, the Gemelli IRCCS Polyclinic Foundation has defined a pathway for the frail patient, defining it as a special category of patient, which takes the patient from admission to the emergency room and leads to discharge.

The Geriatric Frailty Network operates in the Gemelli IRCCS Polyclinic Foundation and is configured as specialized for the II level assessment of frail elderly patients. This special patient population, characterized by a high clinical and care complexity, needs to be followed along the entire path from access to the Emergency Department to discharge to ensure better continuity of care and better outcomes [13].

This pathway starts directly from the Emergency Department through the assessment of the patient identified as frail by the Frailty Unit. Once it has been established that the patient needs hospitalization, the patient can be directly admitted to the pertinent department, which is called the Functional Cognitive Unit, or followed during treatment by the Interdepartmental Geriatrician, who, in addition to supporting the attending physicians during hospitalization, is decisive in defining the continuity of care pathway, being in direct contact with the available territorial services.

## 2. Methods

### 2.1. The Frailty Unit: An Emergency Department Geriatrician

The Frailty Unit is a geriatric team that belongs to the Geriatric Frailty Network, active at the Fondazione Policlinico Universitario “Agostino Gemelli” IRCSS Emergency Department for the evaluation of frail elderly patients.

This project was born within the Fondazione Policlinico Gemelli, where the Emergency Department has an annual incidence of over 65 aged patients of 30%, versus a regional average of 20%. In more detail, the Emergency Department of the Fondazione Policlinico Gemelli in 2020 welcomed 73,474 patients. Of these 22,913 were over 65 (31%).

### 2.2. The Geriatric Assessment

The Consultant of the Frailty Unit aims to evaluate frail elderly patients with the typical tools of Geriatric Medicine, to protect psycho-physical well-being and improve the treatment path of these patients by ameliorating the organizational process and contributing to the creation of tailored paths and procedures.

The evaluation of the frail older adults in the Emergency Department by the Geriatrician of the Frailty Unit consists of the following operating methods:

Anamnesis and multidimensional evaluation (Figure 1): in collaboration with the Emergency Department doctor, the Geriatrician uses as evaluation tools the clinical and socio-care anamnesis, the multidimensional geriatric evaluation (VMD) with particular attention to cognitive and functional status, family network, and second-level frailty assessment scales according to the patient’s characteristics (e.g., ADL, IADL, Braden, RASS, bCAM, Charlson Index, CDR, KPS). In addition, the Geriatrician supports the Emergency Department doctor in the evaluation of clinical problems specific to older adults such as polypharmacy, multimorbidity, and geriatric syndromes (risk of falls, delirium, agitation in dementia, pressure ulcers, hypomobility syndrome, etc.) and, if necessary, in establishing the most appropriate diagnostic-therapeutic procedure.

It is important that the frail elderly patient be assessed by an expert in complexity, to be able to establish what the most appropriate intervention might be, which is not always the most interventionist one. Sometimes, the acute condition in the elderly patient only represents the epiphenomenon of drug interactions or an atypical presentation of acute conditions and it could mask a different type of problem from the one hypothesized (e.g., the drowsiness often hides an infection or a dyselectrolytemia, not an acute neurological problem as the emergency doctor is led to think).

The patient and family members should be interviewed to discuss the preferences and goals of care and to make the patient and family members aware of the risks associated with hospitalization. The team should also explain the need for early consideration of possible alternative routes to hospitalization, the difficulties in post-discharge care, social frailty, possible subsequent discharge from the hospital, and the possible remodeling of treatments according to the proportionality of care.

In these patients, the background is an important part of the treatment process, almost as important as the disease itself. Failure to explore all the characteristics of the patient may lead to taking the wrong path and not resolving the problem.

### 2.3. Discharge Plan

Focused on the hospitalization risk of frail older adults, the Frailty Unit, in collaboration with the Emergency Department doctor, is responsible for identifying possible alternative routes to hospitalization, such as:

Home discharge after an informative interview, eventual contact with the general practitioner, and identification of possible local services for the management of the identified problems (e.g., activation of home assistance, specialist home visits, home health services.).

Home discharge with an appointment at the Geriatrics Day Hospital (Figure 2). This path will be offered to patients who do not have acute clinical needs that require hospitalization but who require a short-term clinical and laboratory re-evaluation and a geriatric follow-up in a multidisciplinary setting (e.g., patient with onset of cognitive impairment, agitation in dementia, movement disorders, psychogeriatric disorders, malnutrition, recurrent falls, social fragility, etc.).

Home discharge with activation of palliative care services. This path will be proposed for patients identified in the Emergency Department as candidates for a home or residential palliative care pathway and who do not have clinical acuity requiring hospitalization. This opportunity should be aimed at patients with diseases that can no longer be cured, but who have symptoms that require various health professionals’ active and constant intervention.

Return to a nursing home. This path will be proposed for patients from health facilities (i.e., non-hospital facilities) after an informative interview with the patient and the family, possible contact with the nursing home in question, with the general practitioner, and identification of possible territorial services for the management of new problems (e.g., activation of home care, specialist home visits.).

Early identification of patients with disease progression requiring hospitalization, eligible for a palliative care service. The Geriatrician evaluates patients in the Emergency Department suffering from oncological or chronic developmental pathologies in their terminal phase (neurodegenerative, cardiovascular, pneumatological, etc.) to share with the patient and the family the personalized treatment objectives and identify the most appropriate intra- and extra-hospital path.

Early identification of patients with a clinical and care condition that does not require hospitalization in an acute hospital setting, but requires temporary institutionalization in post-acute or long-term care.

Early identification of patients with a high risk of difficult discharge. Patients who present already in the Emergency Department with worse cognitive or functional status, loss of autonomy, no social or family network, etc. In addition to receiving an information interview in the Emergency Department about the possible care continuity paths, they will be reported speedily to the Interdepartmental Geriatrician.

Assignment of eligibility for the Functional-Cognitive Unit (a specific department belonging to the frailty network). Fragile older adults, with high clinical-care complexity, presenting clinical acuity with a high impact on the cognitive sphere or functional status and who need hospitalization, can be evaluated by the Geriatrician of the Frailty Unit for eligibility for hospitalization at the Functional-Cognitive Unit.

## 3. Results and Discussion

### 3.1. The Frailty Unit Experience

From January to December 2020, the Frailty Unit evaluated 1191 subjects (Table 1). Patients were selected both on the basis of CFS value, identified at the first level geriatric assessment, but also at the daily interdisciplinary briefing (emergency doctor, bed management, care management, advanced nurse practitioner, social work) handover using SBAR methodology. 723 of these patients were discharged directly from the emergency department with early identification of the continuity of care pathway.

Out of 723 discharged patients, 90 were sent home through their general practitioner, 183 were discharged with an appointment at the Geriatrics Day Hospital and 372 were sent to continue the diagnostic-therapeutic process in a low-intensity medicine division. For 19 patients, palliative care was undertaken directly from the emergency room, while 59 patients were sent or returned to local residential structures.

Out of 1191 patients evaluated, 468 were hospitalized with an early assessment of frailty that facilitated the discharge process, immediately highlighting the major criticalities related to cognitive, functional, social, and welfare conditions (Figure 3).

The early identification of ‘bed blocker’ patients (i.e., those patients in whom it is possible to predict at an early stage that the discharge process will be long and complicated) made it necessary to open a dedicated department, where the cognitive-functional problems were addressed by an expert team to guarantee the identification of continuity of care path (Figure 4). In 2020, 151 patients were admitted to the cognitive-functional unit with an early assessment by the Frailty Unit.

### 3.2. From the Emergency Department to a ‘New’ Geriatric Ward: The Functional-Cognitive Unit

To guarantee comprehensive and continuous care for the elderly patient who comes to the Policlinico Gemelli from the emergency room to admission, a department has been created in close connection with the Frailty Unit, which is called the Functional Cognitive Unit.

The Functional Cognitive Unit is a geriatric hospital division of the Fondazione Policlinico Gemelli to which patients are referred through the following methods:

From Emergency Department, after the Frailty Unit assessment:based on the geriatric short stay observation model, patients with a clinical path or continuity of care already set up for an assumed short-term hospitalization; in these patients, a short-term hospitalization is often necessary, but it is important to guarantee particular attention from a frailty point of view (for example, to prevent geriatric syndromes typical of hospitalization, such as delirium, or falls.).on the long-term care model, patients with a high degree of complexity or potential bed blockers; in these patients, the geriatric approach is fundamental in the treatment of acute cases, as are knowledge of the territory and the possible services supporting safe discharge.

From the wards, after Interdepartmental Geriatrician evaluation, patients who meet the following criteria:the patient must be a “frail older adult” according to the previously specified criteria, when geriatric expertise can ensure more appropriate care and discharge pathwaythe patient must have been admitted to a surgery or specialized medicine department (e.g., vascular diseases, endocrinology, nephrology, neurology.), but no longer requires the specialist expertise for which it was accessed (e.g., a patient who has already undergone surgery and needs to continue medical treatment and observation)the diagnostic process must have been completed and the specific therapeutic process must have started;the continuity of the territorial care process must have been defined or at least the need for it must have been identified

### 3.3. Frailty Evaluation from the Emergency Department to the Wards: The Interdepartmental Geriatrician

At the time of admission of a frail older adult in hospital, the ward doctor has to fill out the Clinical Frailty Scale, highlighting a fragility alert on the information system. This alert can be transformed into a request for interdepartmental geriatric advice, in the clinical judgment of the care provider.

The Interdepartmental Geriatrician consultant belonging to the Geriatric Frailty Network works within Fondazione Policlinico Gemelli IRCSS for the evaluation of frail older adults admitted to the different hospital divisions. Such patients are those for whom, after the Frailty Unit assessment, an alternative route to hospitalization could not be identified.

The intervention is activated through a specific request on the computer system and takes place within 48 h of sending the request.

Once the request for advice has been received, the Interdepartmental Geriatrician evaluates the patient in the recovery ward. The tools for specialist assessment are listed below:-Clinical history including the reason for the hospitalization, the remote pathological history with an evaluation of active and previous comorbidities, the drug history, and the physiological anamnesis;-Multidimensional evaluation that can make use of second-level evaluation tools:Functional (ADL, IADL, Barthel Index) and cognitive status (spatial and temporal orientation, Six Item screener, MMSE, specific neuropsychological tests)State of supervision and delirium evaluation (RASS, CAM);-Evaluation of the presence of geriatric syndromes (delirium and agitation in dementia, functional deterioration, hypomobility syndrome, risk of falls, malnutrition, pressure injury, urinary incontinence);-Assessment of the social and family network;-Evaluation of the continuity of care outside the hospital;-Interview with the patient and family members.

Based on the reason for requesting advice and data collected in the evaluation phase, the Geriatrician draws up a consultation, explaining the results of the evaluation and proposing any advice regarding the diagnostic-therapeutic process and the continuity of care.

The intervention of the Geriatrician can also take place within the surgical departments for the preoperative and postoperative evaluation of frail elderly patients, always upon request for advice from the doctor responsible for the clinical management of the patient.

As part of the geriatric network active within Fondazione Policlinico Gemelli IRCCS, the Interdepartmental Geriatrician carries out evaluations of patients who, from the surgical or specialist medicine departments, are proposed for an internal transfer to the Functional Cognitive Unit or re-evaluates those patients who have already been assessed by the Frailty Unit in the Eemergency Department and need to be taken care of by the geriatrician but who have been admitted for clinical reasons to other departments.

Finally, the Interdepartmental Geriatrician can evaluate frail elderly patients, discharged at home, to schedule a follow-up appointment at the Geriatrics Day Hospital.

### 3.4. Discussion

The organization of the Emergency Department and its architectural configuration are not always suitable for receiving elderly patients who are affected by psycho-physical distress besides the acute morbid condition reason to access the Emergency Department and other comorbidities. The emergency room has environmental characteristics that can increase the risk of presentation and progression of hypomobility complications in older adults [14] such as pressure ulcers, increased risk of falls, delirium, and infections.

The Emergency Department is structured to allow the early diagnosis and treatment of an acute pathology according to a paradigm focused on the management of a single problem (1 patient = 1 clinical problem). However, this paradigm is far exceeded in the elderly population, resulting in new and difficult clinical and organizational challenges. Physiological, psychological, and cognitive age-related changes [15], along with comorbidities and polypharmacy, make the clinical presentation of many acute diseases atypical and heterogeneous [2]. The physical frailty characterizes the geriatric age and determines a greater severity and a deeper impact of acute pathologies, considering the patients’ multimorbidity and polypharmacy [16]. Often the resolution of clinical acuity does not determine, as in a younger patient, the return to the previous state of health but stabilizes the basic chronic condition at a slightly lower level than the previous one, with not only biological but also functional, psychological and welfare repercussions [17,18,19]. Once the therapeutic diagnostic process of ED has been completed, even in the case of clinical-therapeutic success, often the loss of function related to bedside rest, makes it difficult to organize continuity of care and transfer the patient to subsequent settings, especially at home. This, sometimes, leads to prolonged hospitalization, not for the severity of his/her clinical picture or to perform further diagnostics tests and therapies, but due to the difficulty of discharging the patient. However, it is known that hospitalization of frail older adults is associated with an increase in mortality and morbidity [9].

A frail elderly patient in the Emergency Department is generally subjected to a greater number of diagnostic tests, waiting on a stretcher for a longer time to organize the program following urgent treatment. This situation implies a greater consumption of resources and a high risk of delirium and functional decline. It has been documented that management in the Emergency Department of major emergencies such as cardiac arrest, polytrauma, or septic shock is simpler than a single pathology such as urinary tract infection in an older adult, multimorbid, affected by a cognitive impairment complicated by delirium, bed rest syndrome and without family network [20]. In fact, in these types of patients, it is not enough to focus on the acute clinical reason for access, but it is necessary to make a multidimensional geriatric assessment, obviously adapted to the fastest and most pragmatic reality of the emergency room.

The management of the elderly patient in the emergency area would require a multidisciplinary approach for several reasons, with the intervention of a team of experts in geriatric problems, which is lacking in most Italian hospitals at present.

Recent studies show that a geriatric assessment carried out in the emergency department can lead, when possible, to direct discharge from the emergency department, but also to a reduction of the length of stay in case of hospitalization [21]. Other studies are ongoing to better understand how to improve the quality of Emergency Department care for older adults in the acute care setting [22,23].

Based on these premises, the frail older adult represents a peculiar category for which it is necessary to ensure protection in the care path and a personalized specialist evaluation [13,16,19], also during hospitalization and for the discharge plan.

However, the resources required for the implementation of an in-hospital geriatric assessment network are different in national and international settings. These resources depend on several factors: economic, socio-medical, and programmatic.

However, our reality can be encouraging in the approach and knowledge of the care of the frail elderly patient and consequently in the promotion of structural changes in institutions and health policies.

In addition, the teamwork between a geriatric team and doctors of other specialties gives the latter a greater understanding and awareness of the unique care of the elderly patient.

## 4. Conclusions

The FPG has identified the frail elderly patient as a special category in which it is necessary to guarantee protection in the treatment path and a personalized specialist evaluation.

The geriatric frailty network was born as expertise capable of providing this protection and aims to assist the emergency room and ward doctor in the prevention of the most common geriatric syndromes (e.g., delirium, sarcopenia, pressure ulcers), reduce the number of incongruous hospitalizations and the length of hospitalization, and favor targeted alternative care and assistance paths.

## Figures and Tables

**Figure 1 diagnostics-12-01228-f001:**
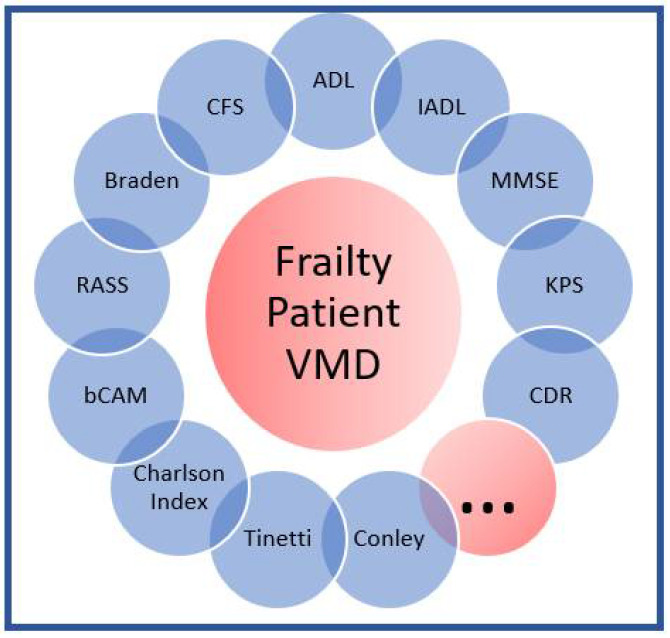
Multidimensional Evaluation.

**Figure 2 diagnostics-12-01228-f002:**
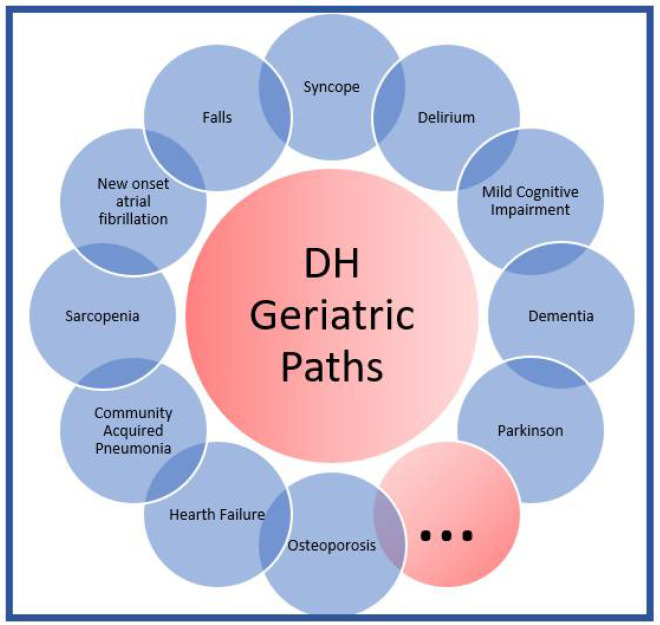
Day Hospital Geriatric Paths.

**Figure 3 diagnostics-12-01228-f003:**
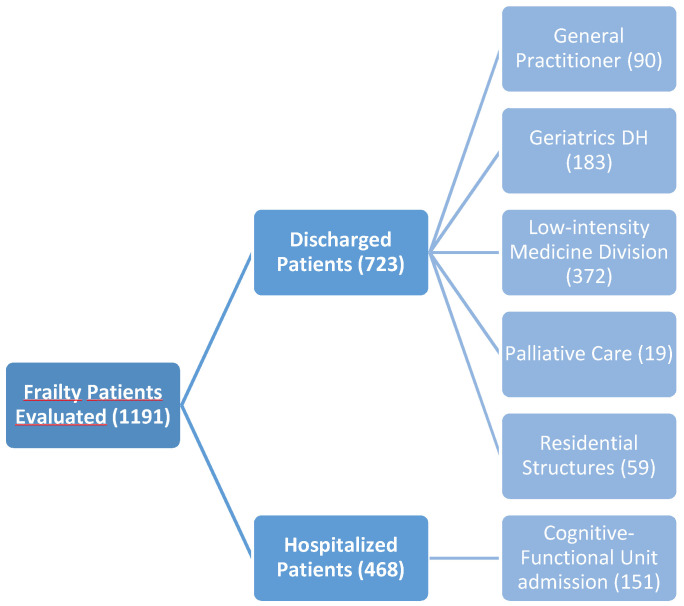
The Frailty Unit Experience.

**Figure 4 diagnostics-12-01228-f004:**
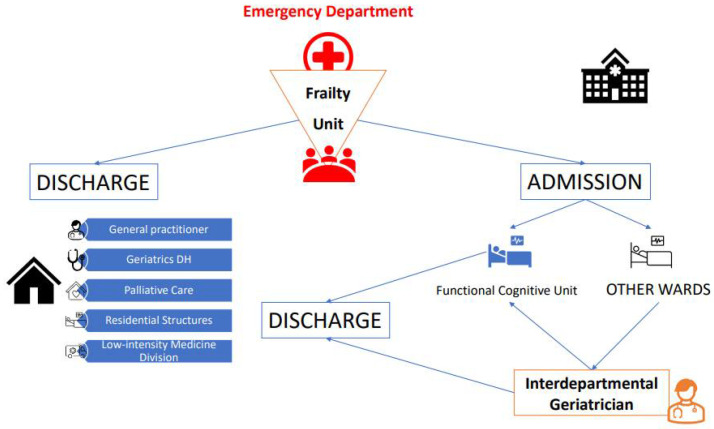
The Geriatric Frailty Network: the discharge flow chart.

**Table 1 diagnostics-12-01228-t001:** Population demographics of patients admitted to the Emergency Department and evaluated by the Frailty Unit. Comparison of discharged and hospitalized patients.

Variable	All *n* 1191	Discharged *n* 723	Hospitalized *n* 468	*p*
Age	82 (74–87)	81 (73–87)	83 (76–86)	<0.001
Sex (Male) Clinical Frailty Scale	516 (45.1) 7 (5–8)	317 (46.6) 6 (4–7)	199 (42.8) 7 (6–8)	0.202 <0.001
* **Emergency Department Presentation** *				
Delirium	135 (11.8)	62 (9.1)	73 (15.7)	<0.001
Dyspnea	251 (21.9)	148 (21.8)	103 (22.2)	0.877
Fever	439 (38.3)	266 (39.1)	173 (37.2)	0.513
Chest pain	50 (4.4)	36 (5.1)	15 (3.2)	0.118
Syncope	107 (9.3)	67 (9.9)	40 (8.6)	0.475
Abdominal pain	90 (7.9)	56 (8.2)	34 (7.3)	0.569
Diarrhea Constipation	35 (3.1) 27 (2.4)	18 (2.6) 15 (2.2)	17 (3.7) 12 (2.6)	0.330 0.681
Dizziness	19 (1.7)	14 (2.1)	5 (1.1)	0.201
Neurological symptoms	68 (5.9)	35 (5.1)	33 (7.1)	0.170
Malaise/fatigue	145 (12.7)	87 (12.8)	58 (12.5)	0.873
* **Clinical History** *				
Charlson Comorbidity Index	4 (2–5)	4 (3–5)	3 (1–5)	<0.001
Ischemic heart disease	190 (16.6)	123 (18.1)	67 (14.4)	0.100
Congestive heart failure	274 (23.9)	129 (19.0)	145 (31.2)	<0.001
Peripheral vascular disease	66 (5.8)	6 (0.9)	60 (12.9)	<0.001
Previous stroke Dementia	125 (10.9) 214 (18.7)	36 (5.3) 58 (8.5)	89 (19.1) 156 (33.5)	<0.001 <0.001
COPD	127 (11.1)	56 (8.2)	71 (15.3)	<0.001
Diabetes	204 (17.8)	96 (14.1)	108 (23.6)	<0.001
Liver chronic disease	36 (3.1)	6 (0.9)	30 (6.5)	<0.001
Rheumatologic disease	15 (1.3)	6 (0.9)	9 (1.9)	0.124
Chronic kidney disease	134 (11.7)	31 (4.6)	103 (22.2)	<0.001
Malignancy	135 (11.8)	41 (6.0)	94 (20.2)	<0.001

## Data Availability

Not Applicable.

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
