# Peer review of "Frailty Network in an Acute Care Setting: The New Perspective for Frail Older People"

_diagnostics, 2022, doi:10.3390/diagnostics12051228_

Round 1

Reviewer 1 Report

With this article the authors adress a very relevant topic.
To improve the relevance of this work I would encourage the authors to present more detailed data to the readers e.g. comparing the group of discharged vs hospitalized patients, also for age subgroups, main problems with frequency.
In the methods of this retrospective study there is some information lacking:
how and by whom was the data collection performed? was it a consecutive patient cohort (all pts. >65y in 2020? or only some of them and how were they selected?)
The "results and discussion" section mainly describes established processes.
Maybe these could be shown in form of a table or flow-chart and some more actual data included in the text. Ideally comparison prior to impementation of the frailty network to with frailty network in place.
Also the work would benefit from including an actual discussion.
Should the pts. qualifying for review by the frailty unit solely be decided by age ? Are there potential other factors or risk scores that could be applied?
How could this process be transferred to external hospitals ?
What is the potential impact ?

Author Response

Dear colleague,

thank you very much for your comments and advice on our work. I will try to answer you point by point, as requested by the editor.

1) You will find in the attached table (Table 1) a demographic description of the sample, with particular attention to emergency department presentation, frailty condition (CFS) and comorbidities of evaluated patients.

2) The data were collected in a database by geriatricians working in the emergency department during the clinical evaluation. Patients were selected as indicated in the commentary on line 202, over a time period of one year.

3) We included, as recommended, a flow chart (line 231) describing the established process.

We would like to perform a comparison prior to impementation of the frailty network to with frailty network in place, however such a comparison would be burdened with a number of biases derives, for example, from the fact that the frailty unit service does not currently operate at night and on holidays, so it is not evenly distributed over 24 hours.

We have also added a new paragraph (3.4 line 306) for further discussion. Within this paragraph, you will also find the answer to point number 5.

4) Evaluated patients are not selected on the basis of age alone. As indicated in the comment to line 200, patients are selected according to CFS value (so as to select more fragile patients) or identified during the handover process at the morning multidisciplinary briefing.

Best regards

Reviewer 2 Report

In the paper, a standard procedure is proposed for the frail older people,called 'The Geriatric Frailty Network'. In my opinion, such research project is interesting and worth considering. However, this paper needs some modifications before consideration for publication. The main issues are listed below.

  1. A few punctuation and grammatical errors are witnessed in the manuscript, better to cross verify.
  2. In the introduction part, the importance of the Geriatric Frailty Network should be expounded moderately.
  3. In Section 2, the review of related works is not enough. There are some recent related works have not described in the manuscript.
  4. Personalized medicine is not defined properly in the introduction. The authors need to revise this part so they keyword in the title is elaborated.
  5. In section 3, it is suggested to add structured tables or figures for better presentation.
  6. The future research directions are suggested to be added in the conclusions.

Author Response

Dear colleague,

thank you very much for your comments and advice on our work. I will try to answer you point by point, as requested by the editor.

1) Thank you for your report. We apologize for the errors, we have carried out a grammatical revision. Now they should be fine.

2) We have revised the introduction, indeed some statements are more suitable for discussion (see section 3.4).

3) We have updated the bibliography, thank you for your suggestion.

4) We agree with your suggestion, we have eliminated 'personalized medicine' from the keywords, as it does not fit perfectly with the concepts we wanted to express.

5) As suggested, we added a table (Table 1) as a demographic description of the sample, with particular attention to emergency department presentation and comorbidities of evaluated patients, and a flow chart describing the established process.

6) We have specified better in the concluding paragraph suggested research directions

Best regards

Round 2

Reviewer 1 Report

The inserted discussion improves the paper

New figure is lacking a title and figure legends

What is the difference between new figure and figure 1 ?

Using less abbreviations may make easier for the reader

Author Response

Dear colleague,
thank you for the kind comments.
As suggested, we revised the images and decided to delete figure one, keeping the flow chart (current figure 4), adding the title.
We have also revised the text and tried to eliminate as many abbreviations as possible.
Thank you very much.